# Depth from a Single Image by Harmonizing Overcomplete Local Network Predictions

**Ayan Chakrabarti**
TTI-Chicago
Chicago, IL
ayanc@ttic.edu

**Jingyu Shao**
Dept. of Statistics, UCLA*
Los Angeles, CA
shaojy15@ucla.edu

**Gregory Shakhnarovich**
TTI-Chicago
Chicago, IL
gregory@ttic.edu

## Abstract

A single color image can contain many cues informative towards different aspects of local geometric structure. We approach the problem of monocular depth estimation by using a neural network to produce a mid-level representation that summarizes these cues. This network is trained to characterize local scene geometry by predicting, at every image location, depth derivatives of different orders, orientations and scales. However, instead of a single estimate for each derivative, the network outputs probability distributions that allow it to express confidence about some coefficients, and ambiguity about others. Scene depth is then estimated by harmonizing this overcomplete set of network predictions, using a globalization procedure that finds a single consistent depth map that best matches all the local derivative distributions. We demonstrate the efficacy of this approach through evaluation on the NYU v2 depth data set.

## 1 Introduction

In this paper, we consider the task of monocular depth estimation—*i.e.*, recovering scene depth from a single color image. Knowledge of a scene's three-dimensional (3D) geometry can be useful in reasoning about its composition, and therefore measurements from depth sensors are often used to augment image data for inference in many vision, robotics, and graphics tasks. However, the human visual system can clearly form at least an approximate estimate of depth in the absence of stereo and parallax cues—*e.g.*, from two-dimensional photographs—and it is desirable to replicate this ability computationally. Depth information inferred from monocular images can serve as a useful proxy when explicit depth measurements are unavailable, and be used to refine these measurements where they are noisy or ambiguous.

The 3D co-ordinates of a surface imaged by a perspective camera are physically ambiguous along a ray passing through the camera center. However, a natural image often contains multiple cues that can indicate aspects of the scene's underlying geometry. For example, the projected scale of a familiar object of known size indicates how far it is; foreshortening of regular textures provide information about surface orientation; gradients due to shading indicate both orientation and curvature; strong edges and corners can correspond to convex or concave depth boundaries; and occluding contours or the relative position of key landmarks can be used to deduce the coarse geometry of an object or the whole scene. While a given image may be rich in such geometric cues, it is important to note that these cues are present in different image regions, and each indicates a different aspect of 3D structure.

We propose a neural network-based approach to monocular depth estimation that explicitly leverages this intuition. Prior neural methods have largely sought to directly regress to depth [1, 2]—with some additionally making predictions about smoothness across adjacent regions [4], or predicting relative

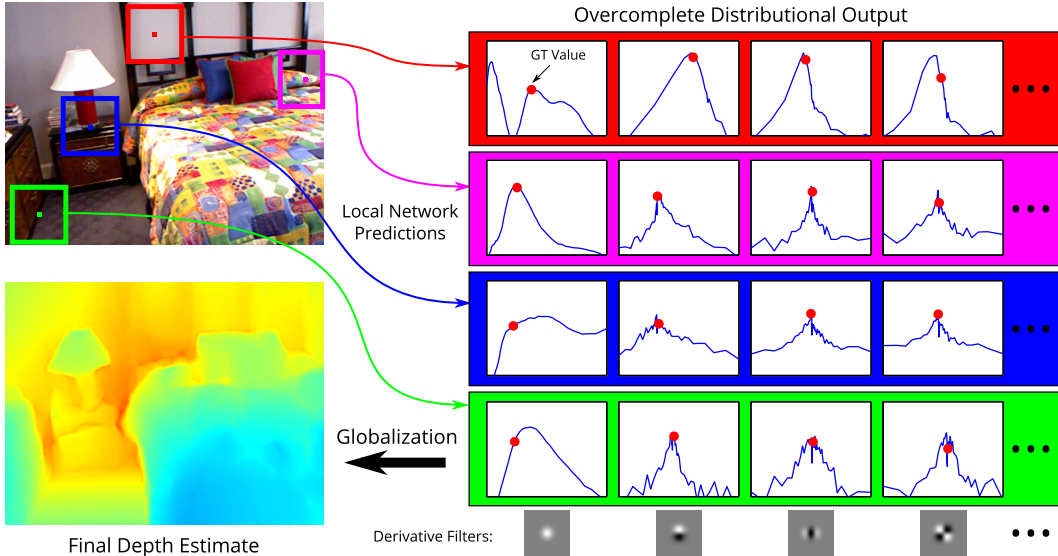

Figure 1: To recover depth from a single image, we first use a neural network trained to characterize local depth structure. This network produces distributions for values of various depth derivatives—of different orders, at multiple scales and orientations—at every pixel, using global scene features and those from a centered local image patch (top left). A distributional output allows the network to determine different derivatives at different locations with different degrees of certainty (right). An efficient globalization algorithm is then used to produce a single consistent depth map estimate.

depth ordering between pairs of image points [7]. In contrast, we train a neural network with a rich distributional output space. Our network characterizes various aspects of the local geometric structure by predicting values of a number of derivatives of the depth map—at various scales, orientations, and of different orders (including the $0^{th}$ derivative, *i.e.*, the depth itself)—at every image location.

However, as mentioned above, we expect different image regions to contain cues informative towards different aspects of surface depth. Therefore, instead of over-committing to a single value, our network outputs parameterized distributions for each derivative, allowing it to effectively characterize the ambiguity in its predictions. The full output of our network is then this set of multiple distributions at each location, characterizing coefficients in effectively an overcomplete representation of the depth map. To recover the depth map itself, we employ an efficient globalization procedure to find the single consistent depth map that best agrees with this set of local distributions.

We evaluate our approach on the NYUv2 depth data set [11], and find that it achieves state-of-the-art performance. Beyond the benefits to the monocular depth estimation task itself, the success of our approach suggests that our network can serve as a useful way to incorporate monocular cues in more general depth estimation settings—*e.g.*, when sparse or noisy depth measurements are available. Since the output of our network is distributional, it can be easily combined with partial depth cues from other sources within a common globalization framework. Moreover, we expect our general approach—of learning to predict distributions in an overcomplete respresentation followed by globalization—to be useful broadly in tasks that involve recovering other kinds of scene value maps that have rich structure, such as optical or scene flow, surface reflectances, illumination environments, etc.

## 2   Related Work

Interest in monocular depth estimation dates back to the early days of computer vision, with methods that reasoned about geometry from cues such as diffuse shading [12], or contours [13, 14]. However, the last decade has seen accelerated progress on this task [1–10], largely owing to the availability of cheap consumer depth sensors, and consequently, large amounts of depth data for training learning-based methods. Most recent methods are based on training neural networks to map RGB images to geometry [1–7]. Eigen *et al.* [1, 2] set up their network to regress directly to per-pixel depth values, although they provide deeper supervision to their network by requiring an intermediate layer

to explicitly output a coarse depth map. Other methods [3, 4] use conditional random fields (CRFs) to smooth their neural estimates. Moreover, the network in [4] also learns to predict one aspect of depth structure, in the form of the CRF's pairwise potentials.

Some methods are trained to exploit other individual aspects of geometric structure. Wang *et al.* [6] train a neural network to output surface normals instead of depth (Eigen *et al.* [1] do so as well, for a network separately trained for this task). In a novel approach, Zoran *et al.* [7] were able to train a network to predict the relative depth ordering between pairs of points in the image—whether one surface is behind, in front of, or at the same depth as the other. However, their globalization scheme to combine these outputs was able to achieve limited accuracy at estimating actual depth, due to the limited information carried by ordinal pair-wise predictions.

In contrast, our network learns to reason about a more diverse set of structural relationships, by predicting a large number of coefficients at each location. Note that some prior methods [3, 5] also regress to coefficients in some basis instead of to depth values directly. However, their motivation for this is to *reduce* the complexity of the output space, and use basis sets that have much lower dimensionality than the depth map itself. Our approach is different—our predictions are distributions over coefficients in an *overcomplete* representation, motivated by the expectation that our network will be able to precisely characterize only a small subset of the total coefficients in our representation.

Our overall approach is similar to, and indeed motivated by, the recent work of Chakrabarti *et al.* [15], who proposed estimating a scene map (they considered disparity estimation from stereo images) by first using local predictors to produce distributional outputs from many overlapping regions at multiple scales, followed by a globalization step to harmonize these outputs. However, in addition to the fact that we use a neural network to carry out local inference, our approach is different in that inference is not based on imposing a restrictive model (such as planarity) on our local outputs. Instead, we produce independent local distributions for various derivatives of the depth map. Consequently, our globalization method need not explicitly reason about which local predictions are "outliers" with respect to such a model. Moreover, since our coefficients can be related to the global depth map through convolutions, we are able to use Fourier-domain computations for efficient inference.

## 3   Proposed Approach

We formulate our problem as that of estimating a scene map $y(n) \in \mathbb{R}$, which encodes point-wise scene depth, from a single RGB image $x(n) \in \mathbb{R}^3$, where $n \in \mathbb{Z}^2$ indexes location on the image plane. We represent this scene map $y(n)$ in terms of a set of coefficients $\{w_i(n)\}_{i=1}^K$ at each location $n$, corresponding to various spatial derivatives. Specifically, these coefficients are related to the scene map $y(n)$ through convolution with a bank of derivative filters $\{k_i\}_{i=1}^K$, *i.e.*,

$$w_i(n) = (y * k_i)(n). \tag{1}$$

For our task, we define $\{k_i\}$ to be a set of 2D derivative-of-Gaussian filters with standard deviations $2^s$ pixels, for scales $s = \{1, 2, 3\}$. We use the zeroth order derivative (*i.e.*, the Gaussian itself), first order derivatives along eight orientations, as well as second order derivatives—along each of the orientations, and orthogonal orientations (see Fig. 1 for examples). We also use the impulse filter which can be interpreted as the zeroth derivative at scale 0, with the corresponding coefficients $w_i(n) = y(n)$—this gives us a total of $K = 64$ filters. We normalize the first and second order filters to be unit norm. The zeroth order filters coefficients typically have higher magnitudes, and in practice, we find it useful to normalize them as $\|k_i\|_2 = 1/4$ to obtain a more balanced representation.

To estimate the scene map $y(n)$, we first use a convolutional neural network to output distributions for the coefficients $p(w_i(n))$, for every filter $i$ and location $n$. We choose a parametric form for these distributions $p(\cdot)$, with the network predicting the corresponding parameters for each coefficient. The network is trained to produce these distributions for each set of coefficients $\{w_i(n)\}$ by using as input a local region centered around $n$ in the RGB image $x$. We then form a single consistent estimate of $y(n)$ by solving a global optimization problem that maximizes the likelihood of the different coefficients of $y(n)$ under the distributions provided by our network. We now describe the different components of our approach (which is summarized in Fig. 1)—the parametric form for our local coefficient distributions, the architecture of our neural network, and our globalization method.

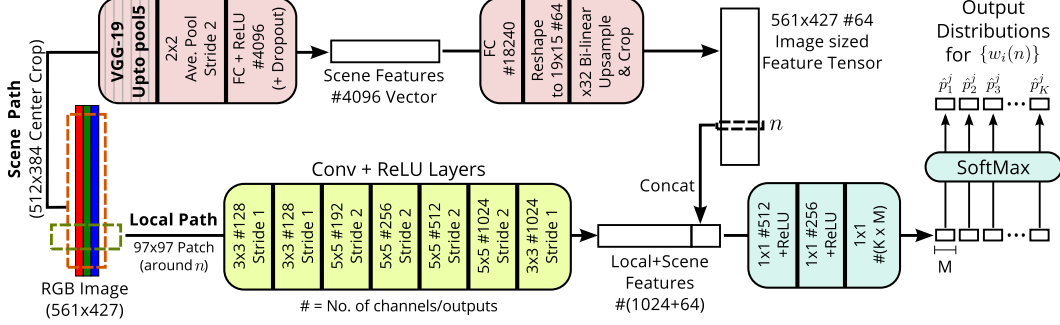

Figure 2: We train a neural network to output distributions for $K$ depth derivatives $\{w_i(n)\}$ at each location $n$, using a color image as input. The distributions are parameterized as Gaussian mixtures, and the network produces the $M$ mixture weights for each coefficient. Our network includes a local path (green) with a cascade of convolution layers to extract features from a $97 \times 97$ patch around each location $n$; and a scene path (red) with pre-trained VGG-19 layers to compute a single scene feature vector. We learn a linear map (with x32 upsampling) from this scene vector to per-location features. The local and scene features are concatenated and used to generate the final distributions (blue).

## 3.1 Parameterizing Local Distributions

Our neural network has to output a distribution, rather than a single estimate, for each coefficient $w_i(n)$. We choose Gaussian mixtures as a convenient parametric form for these distributions:

$$p_{i,n}\left(w_i(n)\right) = \sum_{j=1}^{M} \hat{p}_i^j(n) \frac{1}{\sqrt{2\pi}\sigma_i} \exp\left(-\frac{|w_i(n) - c_i^j|^2}{2\sigma_i^2}\right), \tag{2}$$

where $M$ is the number of mixture components (64 in our implementation), $\sigma_i^2$ is a common variance for all components for derivative $i$, and $\{c_i^j\}$ the individual component means. A distribution for a specific coefficient $w_i(n)$ can then characterized by our neural network by producing the mixture weights $\{\hat{p}_i^j(n)\}$, $\sum_j \hat{p}_i^j(n) = 1$, for each $w_i(n)$ from the scene's RGB image.

Prior to training the network, we fix the means $\{c_i^j\}$ and variances $\{\sigma_i^2\}$ based on a training set of ground truth depth maps. We use one-dimensional K-means clustering on sets of training coefficient values $\{w_i\}$ for each derivative $i$, and set the means $c_i^j$ in (2) above to the cluster centers. We set $\sigma_i^2$ to the average in-cluster variance—however, since these coefficients have heavy-tailed distributions, we compute this average only over clusters with more than a minimum number of assignments.

## 3.2 Neural Network-based Local Predictions

Our method uses a neural network to predict the mixture weights $\hat{p}_i^j(n)$ of the parameterization in (2) from an input color image. We train our network to output $K \times M$ numbers at each pixel location $n$, which we interpret as a set of $M$-dimensional vectors corresponding to the weights $\{\hat{p}_i^j(n)\}_j$, for each of the $K$ distributions of the coefficients $\{w_i(n)\}_i$. This training is done with respect to a loss between the predicted $\hat{p}_i^j(n)$, and the best fit of the parametric form in (2) to the ground truth derivative value $w_i(n)$. Specifically, we define $q_i^j(n)$ in terms of the true $w_i(n)$ as:

$$q_i^j(n) \propto \exp\left(-\frac{|w_i(n) - c_i^j|^2}{2\sigma_i^2}\right), \qquad \sum_j q_i^j(n) = 1, \tag{3}$$

and define the training loss $L$ in terms of the KL-divergence between these vectors $q_i^j(n)$ and the network predictions $\hat{p}_i^j(n)$, weighting the loss for each derivative by its variance $\sigma_i^2$:

$$L = -\frac{1}{NK} \sum_{i,n} \sigma_i^2 \sum_{j=1}^{M} q_i^j(n) \left(\log \hat{p}_i^j(n) - \log q_i^j(n)\right), \tag{4}$$

where $N$ is the total number of locations $n$.

Our network has a fairly high-dimensional output space—corresponding to $K \times M$ numbers, with $(M-1) \times K$ degrees of freedom, at each location $n$. Its architecture, detailed in Fig. 2, uses a cascade of seven convolution layers (each with ReLU activations) to extract a 1024-dimensional *local* feature vector from each $97 \times 97$ local patch in the input image. To further add scene-level semantic context, we include a separate path that extracts a single 4096-dimensional feature vector from the entire image—using pre-trained layers (upto *pool5*) from the VGG-19 [16] network, followed downsampling with averaging by a factor of two, and a fully connected layer with a ReLU activation that is trained with dropout. This global vector is used to derive a 64-dimensional vector for each location $n$—using a learned layer that generates a feature map at a coarser resolution, that is then bi-linearly upsampled by a factor of 32 to yield an image-sized map.

The concatenated local and scene-level features are passed through two more hidden layers (with ReLU activations). The final layer produces the $K \times M$-vector of mixture weights $\hat{p}_i^j(n)$, applying a separate softmax to each of the $M$-dimensional vector $\{p_i^j(n)\}_j$. All layers in the network are learned end-to-end, with the VGG-19 layers finetuned with a reduced learning rate factor of $0.1$ compared to the rest of the network. The local path of the network is applied in a "fully convolutional" way [17] during training and inference, allowing efficient reuse of computations between overlapping patches.

### 3.3 Global Scene Map Estimation

Applying our neural network to a given input image produces a dense set of distributions $p_{i,n}(w_i(n))$ for all derivative coefficients at all locations. We combine these to form a single coherent estimate by finding the scene map $y(n)$ whose coefficients $\{w_i(n)\}$ have high likelihoods under the corresponding distributions $\{p_{i,n}(\cdot)\}$. We do this by optimizing the following objective:

$$y = \arg\max_y \sum_{i,n} \sigma_i^2 \log p_{i,n}\left((k_i * y)(n)\right), \tag{5}$$

where, like in (4), the log-likelihoods for different derivatives are weighted by their variance $\sigma_i^2$.

The objective in (5) is a summation over a large ($K$ times image-size) number of non-convex terms, each of which depends on scene values $y(n)$ at multiple locations $n$ in a local neighborhood—based on the support of filter $k_i$. Despite the apparent complexity of this objective, we find that approximate inference using an alternating minimization algorithm, like in [15], works well in practice. Specifically, we create explicit auxiliary variables $w_i(n)$ for the coefficients, and solve the following modified optimization problem:

$$y = \arg\min_y \min_{\{w_i(n)\}} -\left[\sum_{i,n} \sigma_i^2 \log p_{i,n}\left(w_i(n)\right)\right] + \frac{\beta}{2} \sum_{i,n} \left(w_i(n) - (k_i * y)(n)\right)^2 + \frac{1}{2}\mathcal{R}(y). \tag{6}$$

Note that the second term above forces coefficients of $y(n)$ to be equal to the corresponding auxiliary variables $w_i(n)$, as $\beta \to \infty$. We iteratively compute (6), by alternating between minimizing the objective with respect to $y(n)$ and to $\{w_i(n)\}$, keeping the other fixed, while increasing the value of $\beta$ across iterations.

Note that there is also a third regularization term $\mathcal{R}(y)$ in (6), which we define as

$$\mathcal{R}(y) = \sum_r \sum_n \|(\nabla_r * y)(n)\|^2, \tag{7}$$

using $3 \times 3$ Laplacian filters, at four orientations, for $\{\nabla_r\}$. In practice, this term only affects the computation of $y(n)$ in the initial iterations when the value of $\beta$ is small, and in later iterations is dominated by the values of $w_i(n)$. However, we find that adding this regularization allows us to increase the value of $\beta$ faster, and therefore converge in fewer iterations.

Each step of our alternating minimization can be carried out efficiently. When $y(n)$ fixed, the objective in (6) can be minimized with respect to each coefficient $w_i(n)$ independently as:

$$w_i(n) = \arg\min_w -\log p_{i,n}(w) + \frac{\beta}{2\sigma_i^2}(w - \bar{w}_i(n))^2, \tag{8}$$

where $\bar{w}_i(n) = (k_i * y)(n)$ is the corresponding derivative of the current estimate of $y(n)$. Since $p_{i,n}(\cdot)$ is a mixture of Gaussians, the objective in (8) can also be interpreted as the (scaled) negative log-likelihood of a Gaussian-mixture, with "posterior" mixture means $\bar{w}_i^j(n)$ and weights $\bar{p}_i^j(n)$:

$$\bar{w}_i^j(n) = \frac{c_j^i + \beta \bar{w}_i(n)}{1 + \beta}, \quad \bar{p}_i^j(n) \propto \hat{p}_i^j(n) \exp\left(-\frac{\beta}{\beta+1} \frac{(c_i^j - \bar{w}_i(n))^2}{2\sigma_i^2}\right). \tag{9}$$

While there is no closed form solution to (8), we find that a reasonable approximation is to simply set $w_i(n)$ to the posterior mean value $\bar{w}_i^j(n)$ for which weight $\bar{p}_i^j(n)$ is the highest.

The second step at each iteration involves minimizing (6) with respect to $y$ given the current estimates of $w_i(n)$. This is a simple least-squares minimization given by

$$y = \arg\min_y \beta \sum_{i,n} ((k_i * y)(n) - w(n))^2 + \sum_{r,n} \|(\nabla_r * y)(n)\|^2. \tag{10}$$

Note that since all terms above are related to $y$ by convolutions with different filters, we can carry out this minimization very efficiently in the Fourier domain.

We initialize our iterations by setting $w_i(n)$ simply to the component mean $c_i^j$ for which our predicted weight $\hat{p}_i^j(n)$ is highest. Then, we apply the $y$ and $\{w_i(n)\}$ minimization steps alternatingly, while increasing $\beta$ from $2^{-10}$ to $2^7$, by a factor of $2^{1/8}$ at each iteration.

## 4  Experimental Results

We train and evaluate our method on the NYU v2 depth dataset [11]. To construct our training and validation sets, we adopt the standard practice of using the raw videos corresponding to the training images from the official train/test split. We randomly select 10% of these videos for validation, and use the rest for training our network. Our training set is formed by sub-sampling video frames uniformly, and consists of roughly 56,000 color image-depth map pairs. Monocular depth estimation algorithms are evaluated on their accuracy in the $561 \times 427$ crop of the depth map that contains a valid depth projection (including filled-in areas within this crop). We use the same crop of the color image as input to our algorithm, and train our network accordingly.

We let the scene map $y(n)$ in our formulation correspond to the reciprocal of metric depth, *i.e.*, $y(n) = 1/z(n)$. While other methods use different compressive transform (*e.g.*, [1, 2] regress to $\log z(n)$), our choice is motivated by the fact that points on the image plane are related to their world co-ordinates by a perspective transform. This implies, for example, that in planar regions the first derivatives of $y(n)$ will depend only on surface orientation, and that second derivatives will be zero.

### 4.1  Network Training

We use data augmentation during training, applying random rotations of $\pm 5°$ and horizontal flips simultaneously to images and depth maps, and random contrast changes to images. We use a fully convolutional version of our architecture during training with a stride of 8 pixels, yielding nearly 4000 training patches per image. We train the network using SGD for a total of 14 epochs, using a batch size of only one image and a momentum value of $0.9$. We begin with a learning rate of $0.01$, and reduce it after the $4^{th}$, $8^{th}$, $10^{th}$, $12^{th}$, and $13^{th}$ epochs, each time by a factor of two. This schedule was set by tracking the post-globalization depth accuracy on a validation set.

### 4.2  Evaluation

First, we analyze the informativeness of individual distributional outputs from our neural network. Figure 3 visualizes the accuracy and confidence of the local per-coefficient distributions produced by our network on a typical image. For various derivative filters, we display maps of the absolute error between the true coefficient values $w_i(n)$ and the mean of the corresponding predicted distributions $\{p_{i,n}(\cdot)\}$. Alongside these errors, we also visualize the network's "confidence" in terms of a map of the standard deviations of $\{p_{i,n}(\cdot)\}$. We see that the network makes high confidence predictions for different derivatives in different regions, and that the number of such high confidence predictions is least for zeroth order derivatives. Moreover, we find that all regions with high predicted confidence

Table 1: Effect of Individual Derivatives on Global Estimation Accuracy (on 100 validation images)

| Filters | Lower Better | | | | Higher Better | | |
|---|---|---|---|---|---|---|---|
| | RMSE (lin.) | RMSE(log) | Abs Rel. | Sqr Rel. | $\delta < 1.25$ | $\delta < 1.25^2$ | $\delta < 1.25^3$ |
| Full | 0.6921 | 0.2533 | 0.1887 | 0.1926 | 76.62% | 91.58% | 96.62% |
| Scale 0,1 (All orders) | 0.7471 | 0.2684 | 0.2019 | 0.2411 | 75.33% | 90.90% | 96.28% |
| Scale 0,1,2 (All orders) | 0.7241 | 0.2626 | 0.1967 | 0.2210 | 75.82% | 91.12% | 96.41% |
| Order 0 (All scales) | 0.7971 | 0.2775 | 0.2110 | 0.2735 | 73.64% | 90.40% | 95.99% |
| Order 0,1 (All scales) | 0.6966 | 0.2542 | 0.1894 | 0.1958 | 76.56% | 91.53% | 96.62% |
| Scale 0 (Pointwise Depth) | 0.7424 | 0.2656 | 0.2005 | 0.2177 | 74.50% | 90.66% | 96.30% |

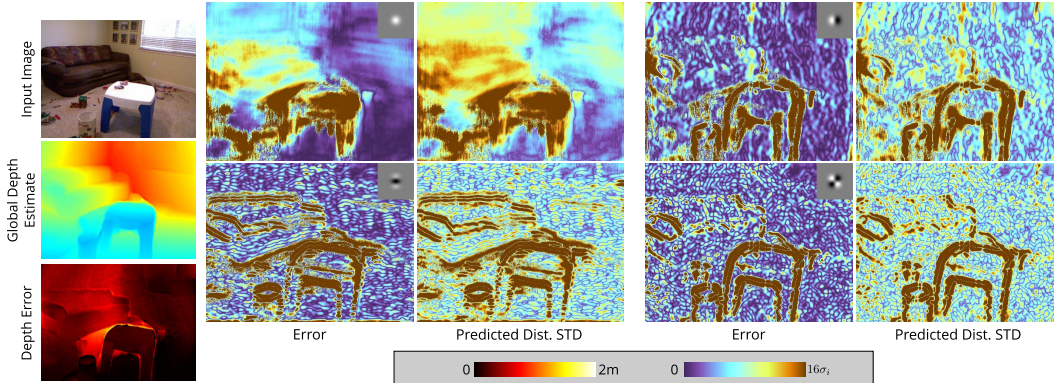

Figure 3: We visualize the informativeness of the local predictions from our network (on an image from the validation set). We show the accuracy and confidence of the predicted distributions for coefficients of different derivative filters (shown inset), in terms of the error between the distribution mean and true coefficient value, and the distribution standard deviation respectively. We find that errors are always low in regions of high confidence (low standard deviation). We also find that despite the fact that individual coefficients have many low-confidence regions, our globalization procedure is able to combine them to produce an accurate depth map.

(*i.e.*, low standard deviation) also have low errors. Figure 3 also displays the corresponding global depth estimates, along with their accuracy relative to the ground truth. We find that despite having large low-confidence regions for individual coefficients, our final depth map is still quite accurate. This suggests that the information from different coefficients' predicted distributions is complementary.

To quantitatively characterize the contribution of the various components of our overcomplete representation, we conduct an ablation study on 100 validation images. With the same trained network, we include different subsets of filter coefficients for global estimation—leaving out either specific derivative orders, or scales—and report their accuracy in Table 1. We use the standard metrics from [2] for accuracy between estimated and true depth values $\hat{z}(n)$ and $z(n)$ across all pixels in all images: root mean square error (RMSE) of both $z$ and $\log z$, mean relative error ($|z(n) - \hat{z}(n)|/z(n)$) and relative square error ($|z(n) - \hat{z}(n)|^2/z(n)$), as well as percentages of pixels with error $\delta = \max(z(n)/\hat{z}(n), \hat{z}(n)/z(n))$ below different thresholds. We find that removing each of these subsets degrades the performance of the global estimation method—with second order derivatives contributing least to final estimation accuracy. Interestingly, combining multiple scales but with only zeroth order derivatives performs worse than using just the point-wise depth distributions.

Finally, we evaluate the performance of our method on the NYU v2 test set. Table 2 reports the quantitative performance of our method, along with other state-of-the-art approaches over the entire test set, and we find that the proposed method yields superior performance on most metrics. Figure 4 shows example predictions from our approach and that of [1]. We see that our approach is often able to better reproduce local geometric structure in its predictions (desk & chair in column 1, bookshelf in column 4), although it occasionally mis-estimates the relative position of some objects (*e.g.*, globe in column 5). At the same time, it is also usually able to correctly estimate the depth of large and texture-less planar regions (but, see column 6 for an example failure case).

Our overall inference method (network predictions and globalization) takes 24 seconds per-image when using an NVIDIA Titan X GPU. The source code for implementation, along with a pre-trained network model, are available at `http://www.ttic.edu/chakrabarti/mdepth`.

Table 2: Depth Estimation Performance on NYUv2 [11] Test Set

| | Lower Better | | | | Higher Better | | |
|---|---|---|---|---|---|---|---|
| Method | RMSE (lin.) | RMSE(log) | Abs Rel. | Sqr Rel. | $\delta < 1.25$ | $\delta < 1.25^2$ | $\delta < 1.25^3$ |
| **Proposed** | 0.620 | 0.205 | 0.149 | 0.118 | 80.6% | 95.8% | 98.7% |
| Eigen 2015 [1] (VGG) | 0.641 | 0.214 | 0.158 | 0.121 | 76.9% | 95.0% | 98.8% |
| Wang [3] | 0.745 | 0.262 | 0.220 | 0.210 | 60.5% | 89.0% | 97.0% |
| Baig [5] | 0.802 | - | 0.241 | - | 61.0% | - | - |
| Eigen 2014 [2] | 0.877 | 0.283 | 0.214 | 0.204 | 61.4% | 88.8% | 97.2% |
| Liu [4] | 0.824 | - | 0.230 | - | 61.4% | 88.3% | 97.1% |
| Zoran [7] | 1.22 | 0.43 | 0.41 | 0.57 | - | - | - |

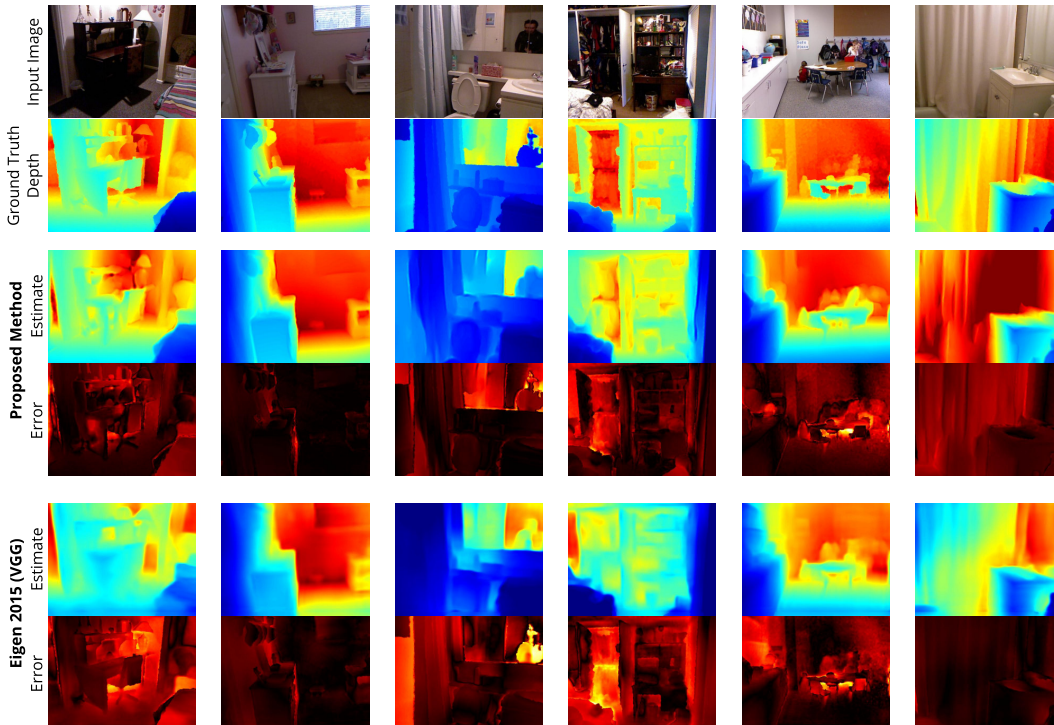

Figure 4: Example depth estimation results on NYU v2 test set.

# 5   Conclusion

In this paper, we described an alternative approach to reasoning about scene geometry from a single image. Instead of formulating the task as a regression to point-wise depth values, we trained a neural network to probabilistically characterize local coefficients of the scene depth map in an overcomplete representation. We showed that these local predictions could then be reconciled to form an estimate of the scene depth map using an efficient globalization procedure. We demonstrated the utility of our approach by evaluating it on the NYU v2 depth benchmark.

Its performance on the monocular depth estimation task suggests that our network's local predictions effectively summarize the depth cues present in a single image. In future work, we will explore how these predictions can be used in other settings—*e.g.*, to aid stereo reconstruction, or improve the quality of measurements from active and passive depth sensors. We are also interested in exploring whether our approach of training a network to make overcomplete probabilistic local predictions can be useful in other applications, such as motion estimation or intrinsic image decomposition.

**Acknowledgments.**   AC acknowledges support for this work from the National Science Foundation under award no. IIS-1618021, and from a gift by Adobe Systems.  AC and GS thank NVIDIA Corporation for donations of Titan X GPUs used in this research.

## Footnotes

*Part of this work was done while JS was a visiting student at TTI-Chicago.

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
