[Reviews · NeurIPS 2016]

Reviewer 1

Summary

This paper presents a method for monocular depth estimation from a single image. A set of derivative filters is convolved with the image producing an overcomplete feature representation. The statistics of this representation is gathered to build a probabilistic model which a neural net attempts to predict at each image location. This allows for an explicit form of uncertainty in the output of the net. After that, all point wise estimated need to combine to form a coherent output map - this is done by using Half Quadratic Optimization. Results on NYU2 are demonstrated and are quite favorable.

Qualitative Assessment

This is a very nice paper. While not groundbreaking, it certainly has a lot of merits. I like the probabilistic nature of the network output, while not an explicit probabilistic model, it still allows to take uncertainty into account. Furthermore, the consensus step where all partial observations are combined is nice as well as it allows for an elegant way of resolving ambiguities. I have several concerns and questions though: 1. Using a fixed set of derivative feature estimates is a nice idea, but I wonder - can these be learned within the same framework? possibly with an iterative algorithm? 2. Section 3.1 - why learn these parameters with K-means? wouldn't EM be a more natural choice with better results? 3. What are typical running times at test time? seems to me this would be quite slow. 4. Looking at Table 2 it seems that the most important features are the order filters - differences between their performance and the full model seem negligible. Any comments about this?

Confidence in this Review

2-Confident (read it all; understood it all reasonably well)


Reviewer 2

Summary

This paper proposes an approach to estimate depth from single images. In contrast to end-to-end CNN methods that directly regress the depth, this paper trains a neural network to predict distributional parameters over the zero-th, first and second order derivatives of the depth map. At test, given the estimate of these distributional parameters for all pixels of the image, a globalization step that ‘harmonizes’ all distributions is proposed. This globalization step aims at estimating the depth map and its corresponding derivatives that maximize the likelihoods of the parameter distributions. The method is evaluated on the NYU v2 depth dataset achieving better results than competing approaches only trained on this dataset, and performing slightly worse that other methods required additional supervision (pretrained VGG network or the outcome of a semantic segmentation).

Qualitative Assessment

This is a technically sound paper. Using CNNs to regress distributional parameters that are later used to estimate depth is an interesting contribution. While the underlying idea is related to that proposed in [Chakrabarti et. al CVPR 2015], I believe there are sufficient differences between the two works to grant a new publication. My main concern is that looking at Figure 3, the estimated individual coefficients seem to have many low-confidence regions. I presume this is because the network is independently applied to local regions of the image. The final accuracy of the estimated depth map, therefore, heavily relies on the globalization procedure which is applied in a final and independent stage. This strategy seems to be in disadvantage with competing approaches such as that of Eigen and Fergus [ICCV 2015], in which several tasks (depth and normal prediction and semantic labeling) are simultaneously addressed by a single network architecture. In a similar way, I wonder if, for the approach proposed in this paper, the local estimation and global harmonization could also be performed by a single network. I would like the authors to further comment on the errors depicted in Fig. 3 and the particularities of the patterns that are drawn. For instance, I understand that the smooth elliptical shapes are the result of the GMMs, but I do not understand the reason of the vertical high frequency stripes.

Confidence in this Review

2-Confident (read it all; understood it all reasonably well)


Reviewer 3

Summary

Given a single RGB image, the proposed approach tries to estimate its depth image exploiting visual cues. The idea is to let a neural network predict a distribution for high order derivatives of the depth for each patch. These retrieved parametrized distributions (GMMs) are used in a global optimization step to construct the depth map itself.

Qualitative Assessment

While the paper is well structured and easy to follow until Section 3.1, there are some open questions on the technical side and the motivation behind the proposed steps. motivation ------------ The authors give no reason for chaining these multiple steps despite their experimental results. Why is a specific intermediate GMM representation needed instead of letting a neural network do what it is good for: learning good intermediate representations? Especially fixing the derivative filters of the first network seems an unnecessary restriction. Why not learning them? The approach has also some similarity to using VLAD/FisherVector features as inputs or the more recently proposed NetVLAD neural network architecture. So what is the difference of the presented approach to the ones mentioned? At least, if there is a distribution representation of intermediate values, I would have expected to see some kind of variance estimate of the depth map. Is this possible? That would be a great plus and good motivation for the intermediate distribution representation. technical points ----------------- - L. 123, what about the other clusters? Did the authors use some standard techniques like splitting highly populated clusters to fill negligible clusters? How many clusters are effectively used? All 64? - How does the number of clusters effect the results? - L. 116, isn't this just linear regression, because everything is fixed but \hat{p}. This seems to be more or less just a normalization step. What happens if one directly uses w_i? - L. 129, why this architecture? - What is the entire runtime during the inference phase, because the first part seems to process patches only? - There are no visual comparisons to any other method. It is hard to judge in which parts the algorithm is more robust than others. - L. 227, it is a no-go to let a reader look up used notations by referring to other papers. - The paper doesn't discuss any failure cases. Are there some failure cases the authors are aware of? ----------------- Thanks for addressing all raised questions in the rebuttal. After reading the authors' rebuttal, I uprated my voting.

Confidence in this Review

2-Confident (read it all; understood it all reasonably well)


Reviewer 4

Summary

The proposed approach uses ConvNets to predict local depth outputs and applies a globalization to predict a globally consistent depth prediction. The authors use a ConvNet to predict weights for a set coefficients, which are represented via a mixture of Univariate Gaussians. These coefficients are directly related to the depth via a convolution operation with set of kernels. The kernels here are derivatives of a Gaussian. Optimization for the coefficients and depth is done in an alternate fashion: fixing the predicted depth they optimize for the coefficients, then fixing the coefficients they optimize for the depth, and so on. - The approach makes no assumptions about local outputs. Other similar local-global methods like, Chakrabarti et. al [2] impose planarity assumption.

Qualitative Assessment

Clarity: + The paper structure is good. + Explanations are presented in a simple language. It was understandable. Novelty: + They idea of using ConvNets for local output prediction is new. - The basic idea of making local prediction and harmonising local outputs to get a consistent global estimation is not new. The same idea is used in several papers, including in Chakrabarti et. al. [2]. Therefore, novelty, is quite limited, especially for NIPS standard. Experiments and Test: - The results in the NYU v2 Dataset are NOT superior to state-of the art methods. They are only comparable. Impact: + The idea of using ConvNets in the area (Globalisation of local outputs) could come to light because of this work. This might help other authors experiment the idea to other problems too.

Confidence in this Review

2-Confident (read it all; understood it all reasonably well)


Reviewer 5

Summary

This paper presents a neural network system that estimates depth from monocular images. The proposed depth estimation system is a cascaded two-stage system. First, the system generates distributions for values of various depth derivatives of different orders, at multiple scales and orientations. Then the system combines these local distributions within a globalization framework.

Qualitative Assessment

This paper has a clear presentation and strong results in general, however, this submission does not provide sufficient novel ideas. This paper is not strong enough to be a NIPS paper. 1. The proposed system is very similar to Eigen et al and Liu et al. From the qualitative and quantitative improvements, it is not clear whether the improvements are from the different neural network architecture or the proposed algorithm. I would expect to a more fair comparison between the proposed algorithm and Eigen et al. in the same neural networks settings. 2. Fairness in experiments. Liu et al. is based on super-pixels, while this work is based on sliding window approach. Therefore, I would expect to see another baseline, applying CRF on top of the proposed local neural networks. Authors are encouraged to either provide justification of not using this baseline or provide new experiments in the rebuttal. 3. The proposed algorithm is not very well motivated. I am not sure why the two-stage cascaded architecture clears the ambiguity in the predictions in line 42. 4. There are quite a lot of related references in semantic segmentation are missing in this submission. Although they are not directly doing depth estimation, most of them have involved similar architecture like this paper. Authors are strongly encouraged to update with these relevant works.

Confidence in this Review

2-Confident (read it all; understood it all reasonably well)


Reviewer 6

Summary

This paper proposed a novel monocular depth estimation algorithm. Instead of directly fitting the depth map, the method predicts a series of filter-bank response coefficients of the depth map. In the prediction stage the proposed method predict the filter response distribution and recover the depth through a global energy minimization step.

Qualitative Assessment

In general the proposed method is technically sound. The general idea also makes sense in better capturing high-level statistics. The method seems novel. However, I have some concerns on how to prove it matters and why the proposed method is chosen. The general idea is somewhat similar with some recent work which incorporates loss function over the high-order statistics on many low-level vision tasks such as flow, monocular depth recovery and image generation (e.g. "DisparityNet" Mayer et al. 2016, "Image Generation with Perceptual Loss" Ridgeway et a. 2015, "DeePSiM loss" Dosovitskiy et al. 2015, "Depth Estimation" Eigen et al. 2015). All the methods use loss functions that capture high-level statistics, or so-called perceptual loss through handcrafted (or learnable) filter map responses. Motivated by this I think a baseline is required to convince the readers: instead of fitting filter-bank response then followed by an energy minimization stage to recover the depth, whether we can fit the depth using the following loss function: the coefficient similarity between the filter responses from GT depth and predicted depth. This scheme does not require the time-consuming alternating inference in prediction, but in the meantime captures the high-order statistics through the handcrafted filters. I wonder what is the performance and why the proposed is favored by the author. The author claims the importance of fitting the coefficient of filter-bank responses, but did not to show such importance through experiments. Is this visually more appealing or better preserve 3D geometry than directly fitting depth map? Or does modeling the uncertainty brings diverse prediction? From section 4.2 I can hardly understand why it matters. From the comparison study, it seems even much less important than pretraining a VGG model on ImageNet or incorporating semantic labeling loss. Moreover, there lacks of some foundation why fit high-order depth map statistics directly, rather than the 3D geometry space, and why such a series of handcrafted gaussian filters are used. In fact, it is totally doable that the author could incorporate the inference stage of eq. 6 into the network and conducted learning the filters in an end-to-end manner. In this way the objective of reconstructing depth could also directly utilized. In fact, I also have a doubt that the indirect regression might hurts the performance in terms of RMSE since the model is not optimized to minimize such a loss. Overall I think the current stage of the paper is on the borderline and I am expecting the author's rebuttal to address my concerns.

Confidence in this Review

2-Confident (read it all; understood it all reasonably well)